# Spatial Structure of NanoFAST in the Apo State and in Complex with its Fluorogen HBR-DOM2

**DOI:** 10.3390/ijms231911361

**Published:** 2022-09-26

**Authors:** Vladislav A. Lushpa, Nadezhda S. Baleeva, Sergey A. Goncharuk, Marina V. Goncharuk, Alexander S. Arseniev, Mikhail S. Baranov, Konstantin S. Mineev

**Affiliations:** 1Shemyakin-Ovchinnikov Institute of Bioorganic Chemistry RAS, Moscow 117997, Russia; 2Moscow Institute of Physics and Technology, School of Biological and Medical Physics, Dolgoprudny 141701, Russia; 3Institute of Translational Medicine, Pirogov Russian National Research Medical University, Moscow 117997, Russia

**Keywords:** fluorogen-activating protein, FAST, nanoFAST, spatial structure, dynamics, ligand specificity, fluorogen, binding constant, structure

## Abstract

NanoFAST is a fluorogen-activating protein and can be considered one of the smallest encodable fluorescent tags. Being a shortened variant of another fluorescent tag, FAST, nanoFAST works nicely only with one out of all known FAST ligands. This substantially limits the applicability of this protein. To find the reason for such a behavior, we investigated the spatial structure and dynamics of nanoFAST, both in the apo state and in the complex with its fluorogen molecule, using the solution NMR spectroscopy. We showed that the truncation of FAST did not affect the structure of the remaining part of the protein. Our data suggest that the deleted N-terminus of FAST destabilizes the C-terminal domain in the apo state. While it does not contact the fluorogen directly, it serves as a free energy reservoir that enhances the ligand binding propensity of the protein. The structure of nanoFAST/HBR-DOM2 complex reveals the atomistic details of nanoFAST interactions with the rhodanine-based ligands and explains the ligand specificity. NanoFAST selects ligands with the lowest dissociation constants, 2,5-disubstituted 4-hydroxybenzyldienerhodainines, which allow the non-canonical intermolecular CH–N hydrogen bonding and provide the optimal packing of the ligand within the hydrophobic cavity of the protein.

## 1. Introduction

“Fluorescence-Activating and absorption-Shifting Tag” (FAST) is a popular emerging tool for fluorescent labeling in living cells [1,2,3]. FAST is a fluorogen-activating protein (FAP), i.e., it can bind small molecules, fluorogens, that are themselves dim fluorescent agents but become a hundred times brighter upon the FAP binding. FAPs represent convenient tools in modern biochemistry because they do not require time to maturate, can be switched on and off on demand, and their optical properties can be modified without time-consuming work on protein mutagenesis.

FAST is distinguished from other FAPs for several reasons, including the small size (14 kDa), high overall brightness, and ability to bind a variety of fluorogens with various light absorption and emission properties [4,5]. The initial variant of FAST was found by directed evolution as a modification of the photoactive yellow protein, PYP [6]. In the past five years, there were several FAST modifications suggested, including the protein variants optimized for particular ligands of various colors [7,8,9], split constructs [10], dimeric proteins [7], and promiscuous variants, optimized for the vast classes of fluorogens [5]. Most of the listed FAST homologs were obtained by the random mutagenesis/directed evolution approach. However, the recently determined NMR spatial structure of FAST in complex with one of its fluorogens [11] allowed the rational design of novel FAST variants. In particular, we found optimized proteins mcFAST-Y and mcFAST-L by structure-based rational mutagenesis [12]. These proteins are 20–30% brighter than FAST in complex with the fluorogen of a particular structural group.

Finally, analysis of the FAST structure in the apo form revealed that the first 30 amino acids form two α-helices in the FAST/fluorogen complex but are unstructured in the absence of ligand [11]. Considering that this N-terminal subdomain does not interact directly with fluorogens, we assumed that a truncated variant of FAST should still be active and thus designed the “nanoFAST” tag—the shortest genetically encoded FAP. However, screening a vast library of FAST fluorogens revealed that nanoFAST works nicely only with one out of all known FAST ligands, HBR-DOM2. This constitutes a substantial difference compared to the parent FAST protein and limits the applicability of nanoFAST. In the present work, to find the reason for such a deficiency and to have a starting point for the further optimization of nanoFAST, we report the NMR investigation of nanoFAST, both in the apo state and in complex with its fluorogen molecule.

## 2. Results

In this work, we posed several tasks. First, we resolved the structure of nanoFAST without the ligand to understand the effect of the N-terminal part of FAST on the structure and stability of its C-terminal domain. Next, we determined the structure of the nanoFAST/HBR-DOM2 complex to find the key features of the ligand, which make it active in complex with the FAP, while other ligands do not work. Finally, we characterized the internal mobility of the protein backbone to assess the stability of the protein.

### 2.1. Structure of NanoFAST in the Apo State

The structure of the nanoFAST apo state was determined using the conventional NMR approach and was based on NOE distances, chemical shifts, and J-couplings; the high overall quality of NMR data allowed obtaining the almost complete chemical shift assignment (Appendix A). The statistics of the input data and the resulting set of NMR structures are provided in Table 1. The resulting conformation is, to a high extent, similar to the previously reported structure of the FAST C-terminal subdomain—they could be superimposed with the backbone RMSD as small as 1.67 A (Figure 1A–C). nanoFAST chain forms a five-strand antiparallel β-sheet (G3-L7, Q15-A18, N63-T72, G75-K85, and Y92-K97), an α-helix (Y50-S59, H5, here and below the elements of secondary structure are indexed similarly to the full-length FAST in complex with N871b [11]), one turn of π-helix (F36-V40) and one turn of 3–10 helix (P28-Q30, H4). The conformation is well defined by the experimental data, as follows from the low pairwise backbone RMSD, excluding the loop 19–27, which is disordered. The similarity of FAST and nanoFAST conformation is supported by the analysis of NMR chemical shifts (Appendix A). Both the Cα and amide chemical shifts revealed the strong correlation, the certain differences are observed only for the β-sheet residues of nanoFAST that are close to the deleted N-terminal part of the full-length FAST protein. According to Cα shifts and the obtained structures, the β-sheet conformation is stabilized in nanoFAST compared to FAST for the C-terminal residues of strand B2 (16–19) and is destabilized for the C-terminal residues of strand B5 (96–99). Both regions are in direct contact with the N-terminus of nanoFAST.

While the structures of the two proteins are almost identical, the drastic difference between the behavior of FAST and nanoFAST becomes evident starting from the first view of the NMR spectra of nanoFAST (Appendix A). The spectra of nanoFAST are well resolved and reveal narrow signals. In contrast, full-length FAST spectra are worse and contain multiple broadened resonances, which belong to both the N- and C-terminal domains. The cross-peaks corresponding to the amide groups of the unstructured C-terminus have similar intensities in FAST and nanoFAST; however, the intensities of peaks corresponding to the structured domain are 2–3 times lower for the FAST protein (Appendix A), which implies the strong contribution of slow motions to the transverse relaxation. The listed results suggest that while the presence of the first 26 residues of FAST does not affect the structure of the C-terminal domain, it substantially alters the protein dynamics, inducing slow motions in almost the whole protein. These motions most likely originate from the transient contacts between the N- and C-terminal domains of FAST.

### 2.2. Structure of nanoFAST/HBR-DOM2 Complex

As a second step, we determined the structure of nanoFAST in complex with its fluorogen HBR-DOM2 under the same conditions as the apo form. The quality of NMR spectra was almost perfect (Appendix A), which allowed direct measurement of 58 ligand/protein distances using the isotope-filtered experiments (Appendix A). The resulting structure was close to the conformation of nanoFAST apo form with the major difference in the region of loop 19–27, which became an α-helix (18–24, H3) upon the ligand binding (Figure 2, Table 1). In addition, analysis with the PDBsum server reveals the difference in the position of the first two β-strands: 3–7/15–18 and 2–7/13–16 in the apo and ligand-bound forms of the protein. HBR-DOM2 is forming three intermolecular hydrogen bonds with the side chains of Y16, E20, and W68 residues of nanoFAST, similar to what was observed in the FAST/N871b complex. The hydrophobic fluorogen-binding pocket is formed by the side chains of T24, F36, V40, A41, T44, I70, P71, and V81. These residues are in tight contact with the ligand, and the F36 aromatic ring is involved in the π-stacking interaction with the phenolic ring of HBR-DOM2.

An interesting aspect of the obtained structure is the ionization state of HBR-DOM2. This fluorogen has two chemical groups that can possibly be ionized, the hydroxyl group of the phenolic ring and the amide group of rhodanine. As shown previously, the hydroxyl group of HBR-DOM2 is deprotonated inside nanoFAST [11], which allows the formation of ligand-protein H-bonds, which involve the sidechain oxygens of Y16 and E20 that act as the proton donors. The amide group of rhodanine, in turn, is known to be deprotonated at pH 5.5. However, for the substituted rhodanine derivative placed into the protein cavity, the pKa can be altered. In our hands, we do not observe the amide proton of HBR-DOM2 in the isotope-filtered NMR spectra. Moreover, the adjacent protein groups do not reveal any exchange NOE cross-peaks to water, which is expected in case the HBR-DOM2 amide proton is present but is broadened due to the fast chemical exchange with the solvent. Moreover, we do not indicate any pH-induced chemical shift changes for the HBR-DOM2 signals in the pH range 5.5–7.0, which implies that the actual pKa of the rhodanine group is much lower than 5.5 (Appendix A). Thus, we can state that in the complex with nanoFAST, HBR-DOM2 is deprotonated and negatively charged in both possible positions.

The comparison of the nanoFAST/HBR-DOM2 structure with the previously reported FAST/N871b complex reveals a high similarity. Positions of the ligand phenolic rings with respect to the β-sheets are identical, and the major differences may be found in the conformation of loops connecting the helices H3 and H5. These discrepancies are most likely caused by different structures of fluorogens. HBR-DOM2 has an additional substituent at the phenolic ring (5-methoxy group), while N871b has a pyridine moiety. These two moieties are packed against the R26 (R52 in FAST) sidechain and alter the position of residues 25–27 and an adjacent turn of the 3–10 helix (Figure 3). The second discrepancy occurs in the region of G43. This residue was extremely important in the initial directed evolution experiment that resulted in FAST creation [6]. In the case of the rhodanine-based HBR-DOM2, G43 is packed closely against the negatively charged nitrogen of rhodanine, most likely forming a non-canonical weak CH–N hydrogen bond [13] (Figure 3C). It is noteworthy that a similar non-canonical CH–N contact may be observed for the CδH proton of P71, another residue that was found essential for the FAST functionality. The found discrepancies are supported by the chemical shift analysis: the maximal changes of ^13^Cα, which depend most on the protein secondary structure, are observed for the residues F2, R26, P42, and G43. (Appendix A). To summarize, we can state that ligand-induced conformations of FAST and nanoFAST are highly similar, with some differences found in the proximity of R26 and G43 residues, which are induced by the structural features of particular ligands.

### 2.3. Dynamics of nanoFAST

In the case of a fluorogen-activating protein, the dynamics are not less important than the structure. For that purpose, we measured the ^15^N relaxation parameters of nanoFAST both in the apo state and in complex with the fluorogen (Appendix A). These data were then converted to generalized order parameters (S^2^), describing the amplitudes of ps-ns motions of the protein backbone and Rex and contributions of slow µs-ms motions to the transverse relaxation. A direct comparison of nanoFAST apo and nanoFAST/HBR-DOM2 reveals that the internal mobility patterns are almost identical. The difference is observed only for the residues 20–30: this region is mobile in nanoFAST apo state, as follows from the decreased NOE and S^2^ values, and agrees with the obtained spatial structure. In contrast, the region is rigid in the HBR-DOM2 complex, with some slow motions detected for the residues 18–26 (H3) and 29, which can be explained by the formation of an H3 helix. A comparison of dynamics of nanoFAST/HBR-DOM2 and FAST/N871b complexes [12] also reveals few differences. Mobile “hot spots” in both complexes include the loop between the helix H5 and strand B3 (62–63, numeration according to nanoFAST) and loops between the strands B3/B4 (72–76) and B4/B5 (86–90). Unlike FAST/N871b, in nanoFAST/HBR-DOM2 complex, the loop 44–47 appears stable, but the slow motions are observed in the helix H3 and adjacent regions, which were absent in the parent protein. Localization of these slow motions in the region of helix H3, which undergoes the most significant structural changes upon the ligand binding, suggests that they originate from the transitions between the ligand-bound and ligand-free states of the protein.

## 3. Discussion

To summarize, we provide the structure and dynamics of nanoFAST protein in the apo state and the complex with HBR-DOM2 fluorogens. Together with the previously reported structures of FAST [11], we now have a dataset of four conformations, including the structures with two drastically different ligands. These data can be used to explain the role of the FAST N-terminal domain in the fluorogen-activating properties of the protein and to understand the ligand specificity of nanoFAST. First, we found that the absence of the N-terminal domain in nanoFAST does not significantly affect the structure of the remaining part of the protein, both with and without the fluorogen. While some subtle discrepancies are observed for the fluorogen-bound states of nanoFAST and FAST, they are most likely induced by the individual properties of N871b and HBR-DOM2 compounds. The dynamics of nanoFAST in complex with HBR-DOM2 do not reveal any drastic differences compared to the FAST/N871b pair: the complex is rigid with very short mobile regions located in the interstrand loops. Some slow motions are observed in the helix H3. However, mobility in the µs-ms timescale should not influence the fluorescence of fluorogens; the characteristic fluorescence lifetime is usually in the range of nanoseconds. Such a similarity of spatial structures of both ligand-free and ligand-bound forms suggests that fluorogens active with FAST should also work with nanoFAST, but it is not observed. In fact, out of the vast variety of FAST fluorogens [4,5,8,9,14,15], only the HBR-DOM2 demonstrates the appropriate fluorescence enhancement when studied in a complex with nanoFAST. Considering that the N-terminal domain does not affect the structure of the C-terminal domain of FAST and the dynamics of its ligand-bound state, one could assume that the explanation for such a behavior can be found in the protein dynamics of the apo state and free energy of the conformation. As we discovered, this hypothesis agrees with the NMR data and dissociation constants of protein/ligand complexes. Indeed, in the FAST apo state, the presence of the N-terminal domain substantially destabilizes the dynamics of the C-terminal “core” of the protein compared to nanoFAST. This C-terminal domain undergoes slow motions in the µs-ms timescale in the absence of ligand, which results in substantial line broadenings observed in the NMR spectra of the FAST apo state. These motions are most likely induced by the transient contacts between the C-terminal core and the unfolded N-terminus of FAST. Thus, we can conclude that the ligand binding by FAST is accompanied by two processes that do not take place in the case of nanoFAST, the folding of an N-terminal domain and the stabilization of the C-terminal core dynamics. These processes could result in a significant enthalpic contribution to the ligand binding. Despite the decrease in entropy due to the ordering of the N-domain, this enthalpic contribution appears much greater and stabilizes the fluorogen/FAP complex. This is clearly seen in the example of HBR-DOM2: its dissociation constant with nanoFAST is 0.85 µM, while Kd with FAST is almost two orders of magnitude lower and equals 21 nM, corresponding to the free energy difference of 2.2 kcal/M. Therefore, we can state that unstructured in the apo state N-terminus of FAST enhances the µs-ms dynamics of the C-terminal core of the protein and serves as a free energy reservoir, which can make the ligand binding more favorable without any direct interactions between the N-terminus and the ligand.

In this regard, it is interesting to consider all the fluorogens that can work in pairs with nanoFAST, even poorly, to understand the principles of the ligand specificity of the protein. All the ligands that were to date proposed as FAST fluorogens could be divided into two vast classes: analogs of 4-hydroxy-benzyldiene-rhodainine (HBR) and modifications of 4-hydroxy-benzylidene-imidazolone (HBI), which are similar to GFP chromophore. The structures of ligands that reveal a considerable fluorescence enhancement (>20×) in complex with nanoFAST are shown in Figure 4, and the first view reveals that all the ligands are modifications of HBR. The reported structure of nanoFAST/HBR-DOM2 may explain this kind of specificity. The activity of an FAP/fluorogen pair is constituted by two major factors: the brightness of the ligand and the dissociation constant of the complex. The brightness is determined by the quantum yield. It depends on the motional stability of fluorogens inside the FAP cavity, while the Kd is determined by the presence of favorable pairwise ligand-protein interaction. According to the spatial structures of FAST and nanoFAST with their fluorogens, HBR analogs provide additional ligand-protein interactions—the non-conventional CH–N hydrogen bonds between the G43 (G69) and P71 (P97) of the protein and nitrogen of the rhodanine ring. These interactions stabilize the complex and improve the Kds compared to HBI fluorogens, which contain the methyl group at this position. Indeed, when similar substituents are present on the benzyldiene moiety of the ligand, the Kds of FAST/HBI complexes are generally higher than the ones of FAST/HBR. Thus, the Kds of the 2,5-dimethoxy- substituted variants of HBI (N1036 and N1048) are as large as 0.71 and 0.15 µM (Appendix A) compared to 0.02 µM Kd of HBR-DOM2, their rhodanine analog [11] (Appendix A).

The second observation that could be done looking at Figure 4 is that almost all the active fluorogens are substituted at positions 2 and 5 of the benzyldiene ring, with the most common substituent at position 2 being a methoxy group. In the case of the structure obtained here, the 2-methoxy group is packed against the hydrophobic side chains of V81 (V107 in FAST), while the 5-OM group is in tight contact with residues V40 (V67) and R26 (R52). One could find that such packing provides the most favorable contacts with minimal rearrangements in the nanoFAST structure that need to take place upon the ligand binding (Appendix A). In the case of the nanoFAST/HBR-DOM2, significant changes between the apo and ligand-bound states are observed only for the residues within the disordered loop 18–27, which forms a helix upon ligand binding. In contrast, N871b induces detectable changes in the position of 3/10-helix 28–30 and in the conformation of residues in the region 42–43 (68–69). The assumption that nanoFAST selects the ligand with the most favorable packing and lowest Kd is in agreement with the known Kds measured for HBR-DOM2 (21 nM) [11] and HBR-2,5DM (8 nm) in complex with FAST [16]. These values are substantially lower than were found for several other ligands, like HMBR (130 nM), HBR-3,5DOM (0.97 µM), and N871b (250 nM) [6,14,16]. We additionally measured the FAST Kds of three other nanoFAST ligands, and they all lay in the range of 10–30 nM, supporting our hypothesis (Appendix A). We need to admit that it could happen that some FAST ligands (e.g., HBI-based) can still bind nanoFAST but lose their fluorogenic properties. To check this option, we investigated the fluorescence and dissociation of the nanoFAST/N871b complex. As revealed by the titration experiment, nanoFAST can enhance the fluorescence of N871b; however, only at extremely high protein concentrations (~100 µM, Appendix A). It implies that N871b is bright when bound to nanoFAST, but its binding affinity is extremely weak. Therefore, the selectivity of nanoFAST is explained by its fluorogen binding ability and not by the low quantum yields of fluorogens inside the ligand-binding pocket. To summarize, we can state that nanoFAST is selective to the 2,5-disubstituted HBR-based ligands, which provide the most favorable packing of an FAP/fluorogen complex.

## 4. Materials and Methods

### 4.1. Sample Preparation

FAST and nanoFAST proteins were synthesized as described earlier [11]. Briefly, the proteins were produced in *E.coli* BL21(DE3). To obtain the ^15^N-^13^C-labeled proteins, the ^15^NH_4_Cl and [U-^13^C]-glucose were used as nitrogen and carbon sources in the M9 medium. The cells were resuspended in a lysis buffer (20 mM Tris, pH 8.0, 500 mM NaCl, 20 mM Imidazole, 200 µM PMSF) and disrupted by ultrasonication. The target proteins were purified by IMAC (buffer: 20 mM Tris, pH 8.0, 200 mM NaCl) and SEC (1× PBS buffer) chromatography. For NMR applications, the target proteins were dialyzed against the NMR-buffer (20 mM NaPi, pH 7.0, 20 mM NaCl, 0.01% NaN_3_) overnight at 4 °C and concentrated up to 1 mM by ultrafiltration (10 kDa MWCO, Amicon Ultra).

To study the structure of a nanoFAST/HBR-DOM2 complex, the ligand was added to the protein sample as an aliquot of 50 mM solution in DMSO-d6 (CIL, USA) at 1.1× molar excess. The resulting solution was centrifuged at 5000× *g* for 5 min, and the supernatant was placed into the 5 mm NMR tube.

### 4.2. NMR Spectroscopy

All the NMR spectra of nanoFAST were recorded using the Bruker Avance III 800 MHz NMR spectrometer equipped with the triple resonance cryogenic probe. Experiments were run at 25 °C, if not otherwise stated. Chemical shift assignment was performed in CARA software [17] using the conventional set of triple resonance experiments, recorded in the BEST-TROSY fashion with the sparse sampling of indirect dimensions [18]. Spectra were processed using the qMDD software [19]. (HB)CB(CGCC)H [20] and (H)CCH-COSY [21] experiments were used to assign the chemical shifts of aromatic side chains. ^3^J_CCo_ and ^3^J_NCo_ couplings were measured from the spin-echo difference constant-time HSQC spectra [22,23]. ^3^J_NHβ_ couplings were derived from the cross-peak intensities in a J-quantitative 3D HNHB experiment [24]. To assign the chemical shift of HBR-DOM2, we used the ^13^C,^15^N-double-filtered 2D NOESY experiment [25]. Intramolecular distances were measured in 3D ^1^H^,15^N-NOESY-HSQC and ^1^H,^13^C-NOESY-HSQC (two spectra with 13C offsets set at 48 and 120 ppm, recorded in D_2_O). Intermolecular contacts were observed directly using the ^13^C,^15^N-filtered,^13^C-edited-NOESY-HSQC [26].

### 4.3. Structure Calculation

Spatial structures were calculated using the automated procedure as implemented in CYANA version 3.98 [27]. Intermolecular NOESY peaklist was assigned manually. The dihedral restraints (𝝋, 𝜒_1_) were obtained from the manual analysis of J-couplings and chemical shifts in TALOS-N software [28]. The geometry of the ligand was optimized in Avogadro [29] using the MMF94 force field, and the resulting .pdb file was used to construct a CYANA-format library. Dihedral angles of the ligand were fixed for the spatial structure calculation. MOLMOL [30] and PyMOL software (Schrödinger LLC) were used for 3D visualization. NMR chemical shifts and spatial structures were deposited into the PDB database under access codes 8AO0 and 8AO1.

### 4.4. Intramolecular Mobility

^15^N longitudinal (T1), transverse (T2) relaxation rates were measured using the pseudo-3D HSQC-based experiments with varied relaxation delay at 30 °C using the Bruker Avance 700 MHz spectrometer [31]. Heteronuclear equilibrium ^1^H,^15^N-NOE magnitudes were obtained using the 1H presaturation for 3 s during the recycling delay. Reference and NOE spectra were recorded in the interleaved mode. Relaxation parameters were analyzed using the model-free approach implemented in the TENSOR2 software, assuming the isotropic rotation [32].

### 4.5. Measurement of Dissociation Constants

The affinity constants for complexes [FAST-chromophore] were determined by spectrofluorometric titration of protein by chromophore solutions with various concentrations on the Tecan Infinite 200 Pro M Nano dual mode plate reader. The protein concentration was 0.10 µM. The least squares fit (line) gave the dissociation constants KD presented as mean ± SD (*n* = 3) in Appendix A. The titration experiments were performed at 25 °C in pH 7.4 PBS (pH 7.4, #cat E404-200TABS, Amresco, Solon, Ohio, USA). Fitting was performed using Origin 8.6 software.

## 5. Conclusions

To conclude, here we report the spatial structures of fluorogen-activating protein nanoFAST (a version of another protein, FAST, lacking the N-terminal domain) in a complex with fluorogens and the apo state. We show that the absence of the N-terminal domain does not affect the structure of the remaining part of the protein. The observed differences between the FAST and nanoFAST are explained by the individual properties of the taken fluorogens. Our data suggest that the N-terminus of FAST destabilizes the dynamics of the C-terminal domain in the apo state. While it does not contact the fluorogen directly, it serves as a free energy reservoir that enhances the ligand binding propensity of the protein. The structure of nanoFAST/HBR-DOM2 complex reveals the atomistic details of FAST and nanoFAST interactions with the HBR-type ligands and explains the ligand specificity. nanoFAST selects ligands with the lowest dissociation constants, 2,5-disubstituted HBRs, which allows the non-canonical intermolecular CH–N hydrogen bonding and provides the optimal packing of the ligand within the hydrophobic cavity of the protein.

## Figures and Tables

**Figure 1 ijms-23-11361-f001:**
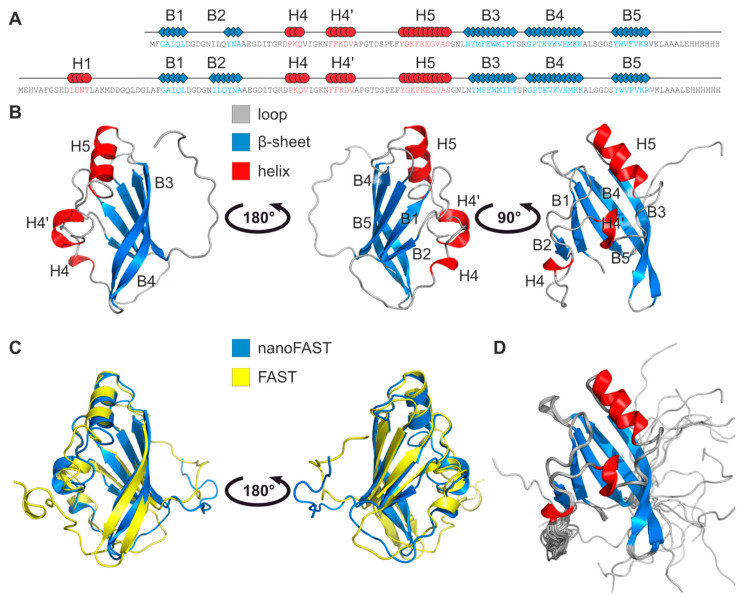
Spatial structure of nanoFAST. (**A**)—Secondary structure of nanoFAST apo state, comparison with Fast. α-helixes are shown in red, β-sheets are shown in blue. H1–H5 and B1–B5 stand for the elements of secondary structure, α-helices and β-strands, respectively. Numbering corresponds to the structure of the ligand-bound full-length FAST protein. (**B**)—Structure of nanoFAST apo state in ribbon representation. (**C**)—Comparison of spatial structures of nanoFAST (blue) and FAST (yellow) in their apo states. (**D**)—Twenty best structures of nanoFAST apo state, superimposed over the backbone atoms of the secondary structure elements. α-Helices are colored in red, β-sheets are shown in blue, coil regions are shown in gray.

**Figure 2 ijms-23-11361-f002:**
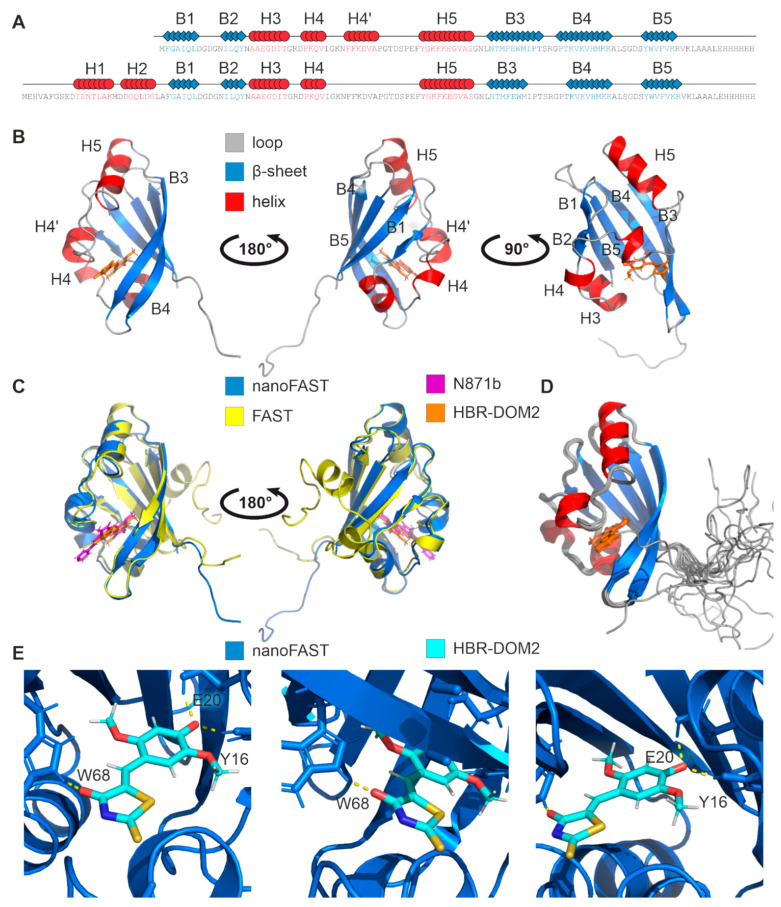
Spatial structure of nanoFAST/HBR-DOM2 complex. (**A**)—Secondary structure of nanoFAST/HBR-DOM2, comparison with FAST/N871b. α-helixes are shown in red, β-sheets are shown in blue. (**B**)—Structure of nanoFAST/HBR-DOM2 in ribbon representation. H1–H5 and B1–B5 stand for the elements of secondary structure, α-helices and β-strands, respectively. Numbering corresponds to the structure of the ligand-bound full-length FAST protein. (**C**)—Comparison of spatial structures of nanoFAST/HBR-DOM2 (blue/orange) and FAST/N871b (yellow/magenta). (**D**)—Twenty best structures of nanoFAST/HBR-DOM2, superimposed over the backbone atoms of the secondary structure elements. α-Helices are colored in red, β-sheets are shown in blue, coil regions are shown in gray. (**E**)—Spatial structures of nanoFAST/HBR-DOM2 with the zoom at the intermolecular hydrogen bonds. Ligand molecule is painted according to the atom type (carbon—cyan, nitrogen—blue, oxygen—red, and sulfur—yellow).

**Figure 3 ijms-23-11361-f003:**
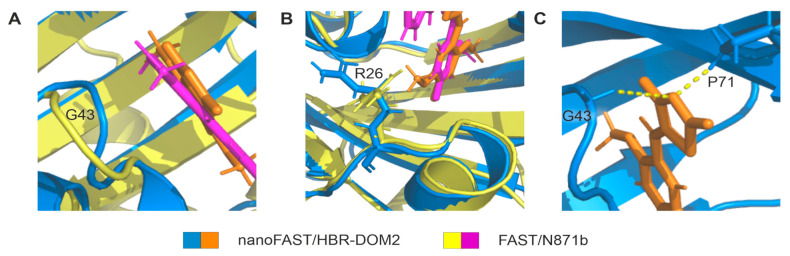
Comparison of nanoFAST/HBR-DOM2 (shown in blue and orange) and FAST/N871b (shown in yellow and magenta). (**A**,**B**)—Superimposed spatial structures of nanoFAST/HBR-DOM2 and FAST/N871b are displayed with the zoom at the regions with the most significant differences, R26 and G43. (**C**)—Non-canonical CH–N contacts between the nitrogen of rhodanine and protons of G43 and P71. Key amino acid residues are indicated by letter and number.

**Figure 4 ijms-23-11361-f004:**
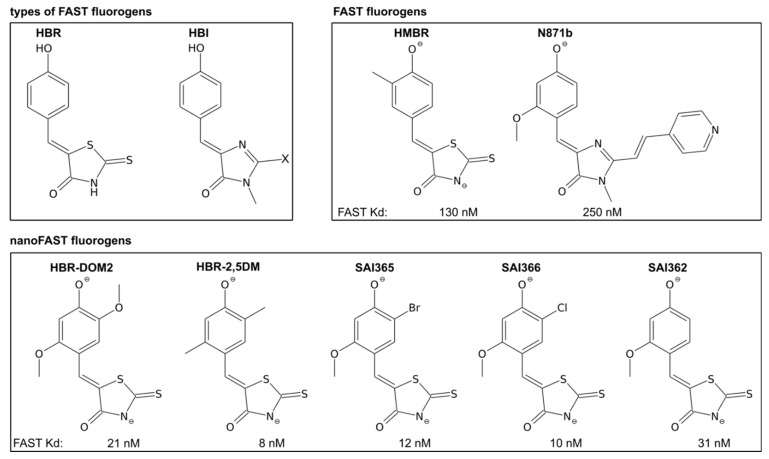
Structures of nanoFAST and FAST fluorogens. Dissociation constants of fluorogens with FAST are provided under the structures, according to the works [5] (HMBR and HBR-2,5DM), [11] (N871b and HBR-DOM2), and according to the current work (SAI362, SAI365, and SAI366).

**Table 1 ijms-23-11361-t001:** Input data and statistics for the obtained spatial structures of nanoFAST and nanoFAST/HBR-DOM2 complex.

Structure	nanoFAST	nanoFAST/HBR-DOM2
NMR distance and dihedral constraints		
Distance constraints		
Total NOE	1994	1390
Intra-residue	459	349
Inter-residue	1535	1041
Sequential (|*i* − *j* | = 1)	527	357
Medium-range (|*i* − *j* | < 4)	339	217
Long-range (|*i* – *j* | > 5)	669	467
Intermolecular	0	58
Hydrogen bonds (upper/lower)	45/45	62/62
Total dihedral angle restraints	199	214
*ϕ*	81	83
ψ	85	88
χ _1_	33	43
Structure statistics		
Violations (mean and SD)		
Distance constraints (Å)	0.0082 ± 0.001	0.0042 ± 0.0008
Dihedral angle constraints (°)	1.052 ± 0.026	1.14 ± 0.015
Max. dihedral angle violation (°)	8.89	11.64
Max. distance violation (Å)	0.32	0.14
Average pairwise r.m.s. deviation (Å), elements of secondary structure ^a^		
Heavy atoms	0.52 ± 0.05	0.79 ± 0.07
Backbone atoms	0.13 ± 0.02	0.38 ± 0.07
Ramachandran analysis (pdbsum)		
most favored regions	75.3%	74.2%
additional allowed regions	23.7%	25.8%
generously allowed regions	1.1%	0%
disallowed regions	0%	0%

^a^ The following residue range was used: 3–8, 11–18, 28–31, 36–38, 42–44, 47–59, 63–71, 76–85, 92–97.

## Data Availability

Spatial structures and NMR chemical shifts for nanoFAST apo state and nanoFAST/HBR-DOM2 complex were deposited into the PDB database under the access codes 8AO1 and 8AO0, respectively.

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
