# Peer review of "Spatial Structure of NanoFAST in the Apo State and in Complex with its Fluorogen HBR-DOM2"

_ijms, 2022, doi:10.3390/ijms231911361_

Round 1

Reviewer 1 Report

The manuscript by Lushpa, et al. determined the structure and measured the dynamics of nanoFAST both in the apo state and in complex with its fluorogen HBR-DOM2 to gain insight as to why nanoFAST only works well with HBR-DOM2 out of all FAST ligands. The authors found that the structures of the apo and ligand-bound nanoFAST are similar to that of their respective FAST parent counterparts. While the structures for the most part are nearly identical, the nanoFAST-ligand complex showed a non-canonical CH-N hydrogen bond between G43 and P71 and the nitrogen atom of rhodanine in HBR-DOM2, that is responsible for the ligand selectivity of nanoFAST. The dynamics of the ligand-bound complexes of nanoFAST and FAST are also fairly identical to each other except for the slow motion observed in helix H3 but that could be attributed to the conformational change upon ligand binding. The authors concluded that the first 26 residues of the FAST in the apo state destabilizes the C-terminal core and that this destabilization is responsible for the wide ligand binding capability of FAST. They also compared the different ligands of nanoFAST and FAST and concluded that nanoFAST prefers fluorogens of the HBR family that is substituted at the 2 and 5 positions.

The results were systematically organized and presented. This manuscript is a good follow-up on the authors’ previous works on FAST and nanoFAST and could potentially guide future optimization of the nanoFAST protein. I just have a few comments or questions that when addressed will help improve the paper:

1. The authors claimed that the 26 residues in the N-terminus of the FAST protein destabilize the C-terminal core and act as a free energy reservoir that will facilitate the binding of the different fluorogens to FAST.

A. Can the authors elaborate more on what they mean by destabilization? Do they mean that the C-terminal domain becomes more dynamic in the slow timescale? Also, what do they mean by acting as a free energy reservoir?

B. They mentioned in the Discussion section that this hypothesis was supported by the NMR data. Are the authors referring to the broad resonances observed in the spectrum of the FAST apo state that became sharper in the nanoFAST apo state? If it is, then this indicates that the C-terminal core is undergoing chemical exchange in the ms-ms timescale in the FAST apo state. I am not sure if the slow motion occurring in this domain counts as destabilization and contributes to enthalpy since the increase in dynamics is an entropic contribution. I think it would benefit the readers if the authors made that section clearer as it is the basis of the main conclusion of the paper.

2. In addition to comparing the dynamics of nanoFAST apo and nanoFAST/HBR-DOM2, the authors also compared the dynamics of the two ligand-bound states: nanoFAST/HBR-DOM2 and FAST/N871b. While the relaxation data of the FAST/N871b are already available in the authors’ previous paper, it will still be helpful for the readers to show the comparison of the relaxation parameters between the two ligand-bound complexes as a supplementary figure so it’s easier to see the mobile “hotspots” in both complexes.

3. In Figure S14, labeling the relevant helices or regions in the zoomed portions of the spatial structures would be helpful.

4. There are some inconsistencies in the labeling of the helices of nanoFAST in the text:

A. Page 2 line 78: P28-Q30, H4 instead of H3?

B. Page 4 line 118: 18-24, H3 instead of H1

C. Page 6 line 162: connecting the helices H1 and H2 (is it really H1 and H2?)

5. In Results section 2.2, aside from showing the spatial alignment of HBR-DOM2 in the structure of the nanoFAST/HBR-DOM2 complex on Figure 2, showing the chemical structure of HBR-DOM2 (or at least referring them to the chemical structure on Figure 4 or S7) when it was first introduced in the text will help orient the readers as the details of the intermolecular interactions were being described.

6. Some typographical errors/improvements:

A. Page 1 line 31: but become hundred times brighter upon the FAP binding.

B. Page 2 line 48: revealed that the first 30 amino acids

C. Page 4 line 120: the position of the first two b-strands: 3-7/15-18 and 2-7/13-16, in the apo and ligand-bound forms of the protein, respectively.

D. Page 8 line 254: …HBI fluorogens,

Author Response

We thank the reviewer for his/her work, we received constructive suggestions and we hope that in the revised version the manuscript was improved. Below we answer the specific comments of the reviewer:

"1. The authors claimed that the 26 residues in the N-terminus of the FAST protein destabilize the C-terminal core and act as a free energy reservoir that will facilitate the binding of the different fluorogens to FAST.

  1. Can the authors elaborate more on what they mean by destabilization? Do they mean that the C-terminal domain becomes more dynamic in the slow timescale? Also, what do they mean by acting as a free energy reservoir?"
  • Indeed, we mean that the presence of the N-terminal domain induces slow motions in the C-terminal core of FAST in the apo state. By acting as a free energy reservoir we imply that the presence of the N-terminus enhances substantially the ligand binding, while this part of the protein does not contact directly with the ligand. Thus, folding of the N-terminal domain, supported by the stabilization of the C-terminal core, provides an additional contribution to the free energy of protein/ligand interaction. We modified the discussion to make it clear.

"B. They mentioned in the Discussion section that this hypothesis was supported by the NMR data. Are the authors referring to the broad resonances observed in the spectrum of the FAST apo state that became sharper in the nanoFAST apo state? If it is, then this indicates that the C-terminal core is undergoing chemical exchange in the ms-ms timescale in the FAST apo state. I am not sure if the slow motion occurring in this domain counts as destabilization and contributes to enthalpy since the increase in dynamics is an entropic contribution. I think it would benefit the readers if the authors made that section clearer as it is the basis of the main conclusion of the paper."

  • We agree with the reviewer and modified the discussion of the paper. We state that the structures of the C-terminal core of FAST and nanoFAST are identical in both the apo and ligand-bound states, and so are the dynamic features of the ligand-bound states. The reason for the observed difference in the fluorogen-activating efficiency of FAST and nanoFAST should therefore be related to the protein dynamics of the apo state and/or to the free energy of the conformation. Indeed, we observe the difference in the dynamics of the apo state that is introduced by the N-terminus, which can be in part responsible for the improved fluorogen binding of FAST. The second factor could be the ligand-induced folding of the N-terminal domain. These two processes, stabilization of the C-terminus and folding of the N-terminus could provide the enthalpy which exceeds the loss of the entropy, and this agrees with the measured dissociation constants. We don't know which of the processes is the major one, so we now provide both as a possible explanation of the enhanced fluorogen-binding ability of FAST. We rewrote the discussion to follow this logic.

"2. In addition to comparing the dynamics of nanoFAST apo and nanoFAST/HBR-DOM2, the authors also compared the dynamics of the two ligand-bound states: nanoFAST/HBR-DOM2 and FAST/N871b. While the relaxation data of the FAST/N871b are already available in the authors’ previous paper, it will still be helpful for the readers to show the comparison of the relaxation parameters between the two ligand-bound complexes as a supplementary figure so it’s easier to see the mobile “hotspots” in both complexes."

  • We added supplementary figure S11, with the comparison of nanoFAST/HBR-DOM2 and FAST/N871b relaxation parameters.

"3. In Figure S14, labeling the relevant helices or regions in the zoomed portions of the spatial structures would be helpful."

  • We modified Figure S14 (S15 in the revised version) and signed the helices. 

"4. There are some inconsistencies in the labeling of the helices of nanoFAST in the text:

  1. Page 2 line 78: P28-Q30, H4 instead of H3?
  2. Page 4 line 118: 18-24, H3 instead of H1
  3. Page 6 line 162: connecting the helices H1 and H2 (is it really H1 and H2?)"
  • We are thankful to the reviewer for the found discrepancies and corrected all the wrong helix labels.

"5. In Results section 2.2, aside from showing the spatial alignment of HBR-DOM2 in the structure of the nanoFAST/HBR-DOM2 complex on Figure 2, showing the chemical structure of HBR-DOM2 (or at least referring them to the chemical structure on Figure 4 or S7) when it was first introduced in the text will help orient the readers as the details of the intermolecular interactions were being described."

  • We did not find a way to add an extra panel to Figure 2 with the chemical structure of HBR-DOM2. However, the structure of HBR-DOM2 can be clearly seen in panel E of Figure 2, to make it readable we introduced the atom-type-based coloring of the ligand. 

"6. Some typographical errors/improvements:

  1. Page 1 line 31: but become hundred times brighter upon the FAP binding.
  2. Page 2 line 48: revealed that the first 30 amino acids
  3. Page 4 line 120: the position of the first two b-strands: 3-7/15-18 and 2-7/13-16, in the apo and ligand-bound forms of the protein, respectively.
  4. Page 8 line 254: …HBI fluorogens,"
  • We corrected all the errors found by the reviewer.

Reviewer 2 Report

Major point:

1)    Line 224/225/235: “we observed that in the FAST apo state, the presence of the N-terminal domain destabilizes substantially the dynamics of the C-terminal "core" of the protein “ 

Is this the case? Although there is line broadening due to exchange, this does not necessarily mean that the C-terminal core itself is destabilized. If the C-terminal core of FAST is dynamic in the apo form, wouldn’t this result in a larger loss of entropy when ligands bind, leading to lower affinity (instead of higher)? Perhaps this can be clarified. If the authors do think that the C-terminal core is destabilized in FAST (or conversely more stable in nanoFAST) this should be supported by thermal or chemical denaturation studies. 

Minor points:

1)    Lines 82: ‘Perfect correlation’, although the correlation is very strong and the data is very convincing, it is not perfect. Strong correlation is more accurate. 

2)    For table 1, the identity of the residues used for r.m.s. correlation should be clearly listed as a footnote. Also, the number of NOEs assigned for the ligand-bound structure should be indicated. 

3)    Could HBI-type ligands still bind to nanoFAST but simply not result in an increase in fluorescence?

Author Response

We thank the reviewer for his/her work, we received constructive suggestions and we hope that in the revised version the manuscript was improved. Below we answer the specific comments of the reviewer:

Major point:

"1)    Line 224/225/235: “we observed that in the FAST apo state, the presence of the N-terminal domain destabilizes substantially the dynamics of the C-terminal "core" of the protein “ 

Is this the case? Although there is line broadening due to exchange, this does not necessarily mean that the C-terminal core itself is destabilized. If the C-terminal core of FAST is dynamic in the apo form, wouldn’t this result in a larger loss of entropy when ligands bind, leading to lower affinity (instead of higher)? Perhaps this can be clarified. If the authors do think that the C-terminal core is destabilized in FAST (or conversely more stable in nanoFAST) this should be supported by thermal or chemical denaturation studies."

  • This part of our discussion caused the questions of both reviewers, for this reason, we rewrote it. Our main thought is that we search for the differences between FAST and nanoFAST that could explain the altered ligand binding efficiency. We show that the structures of the C-terminal core of FAST and nanoFAST are identical in both the apo and ligand-bound states, and so are the dynamic features of the ligand-bound states. Therefore we state that the sought-for difference could be related to the dynamics of FAST in the apo state and/or with the free energy of FAST conformation. We suggest that there are two processes that distinguish the ligand binding in FAST and nanoFAST: the folding of an N-terminal domain and the stabilization of the C-terminal core. Both processes are accompanied by the loss of entropy, however, both processes can provide a favorable enthalpic contribution to the free energy of ligand binding, which could overweight the loss of the conformational entropy. Additionally, we need to say that in our opinion, the enhanced mobility is equivalent to destabilization, because instead of one conformation, corresponding to the energy minimum, we now observe multiple conformations, which implies the changes in the free energy landscape. However, to express our thoughts strictly, in the revised version we now always specify that stabilization/destabilization refers to the protein dynamics and not to thermal stability. 

Minor points:

"1)    Lines 82: ‘Perfect correlation’, although the correlation is very strong and the data is very convincing, it is not perfect. A strong correlation is more accurate. "

  • We modified "perfect" to "strong", as suggested.

"2)    For table 1, the identity of the residues used for r.m.s. correlation should be clearly listed as a footnote. Also, the number of NOEs assigned for the ligand-bound structure should be indicated."

  • We added the number of intermolecular NOEs and a footnote, describing the residue range that was used for RMSD calculation to Table 1 of the revised version.

"3)    Could HBI-type ligands still bind to nanoFAST but simply not result in an increase in fluorescence?"

  • Indeed the HBI-type ligands can still bind to nanoFAST, but lose their fluorogenic activity due to some mobility issues. To check this possibility, we additionally investigated the binding of N871b (an HBI-type ligand)  by nanoFAST. We titrated N871b by nanoFAST up to the concentrations of 100 uM and observed the substantial enhancement of fluorescence at high excess of the protein. It means that nanoFAST is able to enhance the fluorescence of HBI-type ligands, but the binding is very weak and it is possible only at the high excess of the protein (~1000:1 at 0.1 uM ligand concentration). Assuming that the brightness of N871b in nanoFAST and FAST are identical, the obtained binding curve may be approximated by Kd ~1 mM. We added a supplementary Figure and modified the discussion of the manuscript to answer this comment of the reviewer.

Round 2

Reviewer 2 Report

The authors have addressed all of my previous comments. I support publication of this manuscript in IJMS.